# In Silico Screening of Quorum Sensing Inhibitor Candidates Obtained by Chemical Similarity Search

**DOI:** 10.3390/molecules27154887

**Published:** 2022-07-30

**Authors:** Sharath Belenahalli Shekarappa, Hrvoje Rimac, Julian Lee

**Affiliations:** 1Department of Bioinformatics and Life Science, Soongsil University, Seoul 06978, Korea; sharathbs@ssu.ac.kr; 2Department of Pharmaceutical Chemistry, Faculty of Pharmacy and Biochemistry, University of Zagreb, 10000 Zagreb, Croatia; hrvoje.rimac@pharma.unizg.hr

**Keywords:** quorum sensing, AI2 inhibitor, molecular docking, molecular dynamics

## Abstract

Quorum sensing (QS) is a bacterial communication using signal molecules, by which they sense population density of their own species, leading to group behavior such as biofilm formation and virulence. Autoinducer-2 (AI2) is a QS signal molecule universally used by both gram-positive and gram-negative bacteria. Inhibition of QS mediated by AI2 is important for various practical applications, including prevention of gum-disease caused by biofilm formation of oral bacteria. In this research, molecular docking and molecular dynamics (MD) simulations were performed for molecules that are chemically similar to known AI2 inhibitors that might have a potential to be quorum sensing inhibitors. The molecules that form stable complexes with the AI2 receptor protein were found, suggesting that they could be developed as a novel AI2 inhibitors after further in vitro validation. The result suggests that combination of ligand-based drug design and computational methods such as MD simulation, and experimental verification, may lead to development of novel AI inhibitor, with a broad range of practical applications.

## 1. Introduction

Quorum sensing (QS) is chemical communications between bacteria used to sense their own population. QS leads to various group behaviors, including bioluminescence, virulence, and biofilm formation [1]. Gram-positive and gram-negative bacteria use oligopeptides [2,3] and N-acyl homoserine lactone [4,5] for quorum sensing, respectively. In contrast to these two types of signal molecules, autoinducer-2 (AI2) is used by both gram-positive and gram-negative bacteria [6,7,8,9]: it is a universal QS signal molecule that can be used across species [10]. The inter-special nature of the AI2-mediated QS is particularly crucial for biofilm formation on dental gum, because such a biofilm is formed by aggregation of diverse species of oral bacteria [11]. In particular, *Fusobacterium nucleatum* is a major target of AI2 inhibition, because *F. nucleatum* recruits various species of oral bacteria to form biofilm on dental gum, leading to periodontitis. Various AI2 inhibitors for *F. nucleatum* have been found, including furanone compounds [12,13], D-ribose [13,14], and D-galactose [15], where D-galactose has actually been used commercially for prevention of dental gum diseases [16]. However, the AI2 inhibiting activities of these molecules have also been assessed by using *Vibrio harveyi* as the target, because the QS of *V. harveyi* can be easily detected from the resulting bioluminescence. Due to the universal nature of the AI2-mediated QS, a molecule that inhibits QS of *V. harveyi* is also shown to inhibit that of *F. nucleatum* [15]. This fact is useful in the light of the fact that the QS receptor of *F. nucleatum* is still unknown and the structure of *V. harveyi* QS receptor, LuxP, is well known, so that the latter can be used in place of *F. nucleatum* QS receptor for computational purposes. AI inhibitors have also been developed for other bacterial species, such as *V. harveyi* [17,18] and avian pathological *Eschrichia coli* [19]. Inhibitors of AI molecule production have also been developed [20,21]. Nonetheless, it is still desirable to find AI2 receptor inhibiting molecules which are effective in small doses, easy to produce, and are not toxic. In this respect, it is particularly useful to look for QS activities of well-known substances in chemical databases [15,22]. Finding a hitherto unknown QS activities of such substances can lead to the development of novel QS inhibitors by drug-repositioning [23].

An attractive approach for the selection of ligands for computational screening is its chemical and structural similarity with that of the molecules with known bioactivity. In addition, fingerprinting methods such as MACCS, ECFP, path-based fingerprints, and many others are used in characterizing properties of compound collections such as chemical diversity, density in chemical space, and content of biologically active molecules [24,25,26]. Based on these criteria, we selected molecules in chemical database which are chemically similar to known AI2 inhibitors of LuxP, the quorum-sensing receptor of *Vibrio harveyi*, the bacteria often used for experimental test of quorum-sensing inhibition activity of AI2 via detection of bioluminescence. We then performed in silico screening of these molecules using molecular docking and molecular dynamics (MD) simulations. We have found that there are molecules with so far undescribed stable AI2 receptor binding capabilities, which could be developed as novel AI2 inhibitors after further in vitro validation.

## 2. Results

### 2.1. Generation of AI2 Inhibitor Candidate Library

The ligand-based drug discovery (LBDD) approach was applied to construct a library of potential AI2 inhibitors, where the molecules were selected based on 2D molecular similarities/fingerprints with known inhibitors [26]. Due to the universality of the AI2-mediated QS between different bacterial species, it is expected that a molecule that inhibits AI2 binding to the QS receptor of a particular species will have QS inhibitory effect in other species as well. Therefore, we looked for known molecules that bind to LuxP, the QS receptor of *Vibrio harveyi*. We chose 33 known AI2 receptor inhibitors based on activity outcome(active/inactive) and mean activity value (Appendix A), and performed pairwise similarity search against PubChem database. Molecular similarities were evaluated using PubChem fingerprints and Tanimoto coefficient (*T*_c_). Only molecules with *T*_c_ ≥ 0.9 were selected. Since PubChem similarity search returns hits without the similarity score between the query and the hit molecules, we selected those that were found in hit lists for more than one query molecule, under the assumption that such molecules are more likely to be active (true positive) than those that appear in the hit list for a single query. A total of 8196 compounds were selected. We then filtered the compounds based on Lipinski’s descriptors calculation (RO5), where we selected the molecules with molecular weight ≤ 500, xLogP < 5, number of hydrogen bond donors ≤ 5, and number of hydrogen bond acceptors ≤ 10. We extracted a unique representative compound in terms of isomeric SMILES for each group of multiple compounds with the same connectivity (stereoisomers). These additional filters resulted in 2917 unique compounds. Finally, the top 10 hits from the filtered list of 2917 unique compounds were taken based on the frequency of their return (hit frequency), and were then docked to the LuxP structure to assess their binding affinity as AI2 inhibitors. 

### 2.2. Molecular Docking

Before docking the inhibitor candidates to the receptor structure, we first tested the performance of the docking procedure by redocking AI2 to the crystallographic structure of LuxP. We found that the root-mean-square deviation (RMSD) between redocked and co-crystallized ligand structure is low (2.077 Å) and the position and orientation were in excellent agreement with the original structure of the co-crystallized bound complex (Figure 1). Therefore, the docking procedure can be considered to be reliable, and structures obtained by docking could be used as initial structures for MD simulations.

In case of the ligands, for each molecule, conformations to be used as inputs for molecular docking were generated by the ETKDG algorithm, followed by subsequent minimizations using MMFF94s force field. The resulting input structures did not deviate much from each other, with the lowest root-mean-square deviation (RMSD) value being ≤ 2 Å, which is considered to be of high quality [24]. These input conformations were then docked to the LuxP structure using Smina Vinardo. The output from the Vinardo results were further rescored using random forest scoring function (RF-score-v4). Among the top hits, we chose top 5 molecules based on the binding affinity. 1,4-dihydroxypentadecane-2,3-dione (PubChem CID: 91228998) showed the highest binding affinity, followed by (2*R*,3*S*,4*R*)-1,2,3,4-tetrahydroxypentadecan-5-one (PubChem CID: 141428452), 5,8-dihydroxytetradecane-6,7-dione (PubChem CID: 146305585), 1,4-dihydroxytetradecane-2,3-dione (PubChem CID: 90901763), and 1,2-dihydroxytetradecan-3-one (PubChem CID: 144603006). Details of the top five hits are shown in Table 1. The details of interactions between the top five hit molecules in the active site of LuxP are depicted in Figure 2 and Figure 3. All these analyses were performed using Maestro 12.3 (*Schrödinger Release 2022-2*; Maestro, Schrödinger, LLC.: 2021) and Chimera [27,28]. For brevity, these molecules, as well as their complex with LuxP, will be referred by their serial number in Table 1 from here on.

### 2.3. Molecular Dynamics (MD) Simulation and Binding Free Energy Calculation

Further validation of docking results was done by MD simulations and binding free energy calculation. In particular, one can check for possible unstable docking poses [25] and ligand interactions with protein, as well as determine contributions of individual amino acid residues to ligand binding. MD simulations of LuxP complexed with five selected compounds (Table 1) were performed to evaluate the stability and binding energies of the complexes.

To assess stability of the protein–ligand complex, RMSDs of receptor and ligand conformations relative to their initial conformations were monitored for each trajectory at regular time interval of 20 ns. Receptor and ligand RMSDs relative to their initial conformations are shown in Figure 4 (A) and (B), respectively, where RMSDs were computed after the receptor structures are superposed. Among the five complexes, complexes 1 and 5 had the lowest mean receptor backbone RMSD of 1.196 ± 0.58 Å and 1.21± 0.43 Å, respectively, while complex 3 had the highest RMSD value (1.52 ± 0.61 Å). The mean backbone RMSD of all ligands was above 5 Å for the entire 120 ns simulations. 

The root mean square fluctuation (RMSF) is another important parameter that reveals flexibility of each individual residue throughout the simulation and is shown in Figure 5A. Among all the complexes, amino acid residues from 50 to 200 in complex 3 showed the highest flexibility compared to the other complexes. It is interesting to note that amino acids in the ligand binding region of the protein show a certain level of rigidity. This might be an indication that the structure of the protein has been optimized to minimize fluctuations of these residues during the evolution.

Binding free energies (Δ*G*_BIND_) were also evaluated for the five ligand–receptor complexes using the MM-GBSA method [29]. As can be seen in Table 2, complex 1 has relatively weak interactions compared to the other complexes. Complex 5 has the strongest interactions among all the tested ligands, followed by complex 2, both of which are significantly stronger than that of the other complexes. Top ten contributing amino acids of LuxP for the binding free energy of the five complexes are shown in the Table 3. The solvation free energies were evaluated by solving the generalized Born equation, and the total binding free energies were calculated as the sum of various interactions (Table 4). The van der Waals (vdW) interactions have a vital role in binding, while net polar contributions (Δ*G*_GB_) are positive in all five complexes. Among the five complexes, complexes 2 and 5 have the lowest binding free energy and the most favorable van der Waals interaction energy. The number of intermolecular hydrogen bonds for all complexes are shown in Figure 5B, where complexes 4 and 5 show relatively strong intermolecular interactions during the MD simulations. Based on all these analyses, complex 5 (PubChem CID: 141428452, (2R,3S,4R)-1,2,3,4-tetrahydroxypentadecan-5-one) might be the best candidate as a AI2 inhibitor.

## 3. Discussion

In this work, we searched for potential inhibitor candidates of the AI2 receptor by combining ligand-based search with molecular docking and MD simulations. One of the practical applications of AI2 receptor inhibition is prevention of dental periodontitis caused by biofilm formation, since such biofilm is formed by multiple species of bacteria, and AI2 receptor-mediated quorum sensing is a universal means of communication across species. The quorum sensing inhibition without hindering the growth of bacteria is especially valuable, because such a treatment can deter pathogenic behavior of bacteria without inducing drug resistance. Therefore, we deliberately left out any molecules with known anti-bacterial activity from the initial list. The key pathogen causing periodontitis is *F. nucleatum*, which emits AI2 signal to recruit other bacterial species to form a biofilm. Although the quorum sensing receptor of is *F. nucleatum* is assumed to be a galactose-binding protein, its structure is so far unknown. However, utilizing the fact that AI2 is universally used across species and the AI2 receptors of diverse species of bacteria share structural similarity, we used quorum sensing receptor of *V. harveyi*, LuxP, as the target structure for docking and MD simulations. We constructed a library of molecules that share similar chemical structures with known AI2 inhibitors and docked them to the known LuxP structure. Molecular docking revealed that compounds 5,8-dihydroxytetradecane-6,7-dione (PubChem CID: 146305585), 1,4-dihydroxypentadecane-2,3-dione (PubChem CID: 91228998), 1,4-dihydroxytetradecane-2,3-dione (PubChem CID: 90901763), 1,2-dihydroxytetradecan-3-one (PubChem CID: 144603006), and (2R,3S,4R)-1,2,3,4-tetrahydroxypentadecan-5-one (PubChem CID: 141428452) showed the highest binding affinity to the LuxP protein. We then performed additional MD simulation on these five complexes and we found that they indeed form stable complexes with LuxP. These molecules also have good drug-likeness properties, suggesting their potential as AI2 receptor inhibitors. 

Computational methods are valuable for saving time and costs by screening potential inhibitors for further experimental validation. Previously, AI2 inhibiting capability of D-galactose has been found by a method purely based on bioinformatics, where the sequences of known quorum sensing receptors, LuxP of *V. harveyi*, LsrB of *S. typhimurium*, and RbsB of *A. actinomycetemcomitans* were searched against protein sequences of *F. nucleatum* to find that the D-galactose binding protein have a sequence similar to these proteins and hence may function as a quorum-sensing receptor [15]. An alternate approach of using molecular docking and MD simulation have been employed, where a molecule with a high activity as AI-2 inhibitor has been found [22]. The current method is somewhat intermediate between these methods, because although we use molecular docking and MD simulations, we first select the initial candidates based on molecular similarity with known quorum-sensing inhibitors in order to increase the chance of finding a molecule with AI2 inhibiting capabilities. The current method is also somewhat similar to the method used for development of inhibitor of AI2 production [20], but the target of the current method is AI2 reception instead of its production.

The current method is different from that of ref. [21] in that the homology modeling of the unknown receptor structure was not attempted, under the assumption that potential *F. nucleatum* AI2 inhibitor will inhibit the receptor of any other bacterial species. The current method is also different from that of ref. [22], where intermediate conformations between open and closed forms of the quorum-sensing receptors were generated and used for molecular docking and subsequent MD simulations. Only the closed form of LuxP is used in this work for simplicity, which might lead to some bias towards relatively small molecules with a size similar to AI-2. Among the molecules that were not selected by the current protocol due to the low docking scores with the closed conformation of the quorum-sensing receptor, there might have been some molecules that inhibit the AI2 receptor in its more open conformations, which should be examined further. The significance of the current work is to development of a protocol for screening potential AI2 inhibitors among *known* substance, so that the time and cost of synthesizing new molecules. However, it is inevitable that the result of a computational study is limited without experimental validation. Therefore, it goes without saying that the single most important future direction is to combine the current protocol with in vitro tests, such as measurement of *V. harveyi* bioluminescence or biofilm formation by *F. nucleatum*. Once the molecules found in this work are experimental validated, it will be able to be applied to various practical areas such as dental disease prevention.

## 4. Materials and Methods

### 4.1. Preparation of the Ligand-Based Compound Library

The bioactivity data for autoinducer 2-binding periplasmic protein LuxP of *Vibrio harveyi* was retrieved by collecting available target annotations from ChEMBL database (https://www.ebi.ac.uk/chembl/, accessed on 12 May 2022), Binding database (https://www.bindingdb.org/rwd/bind/, accessed on 12 May 2022), and PubChem database (https://pubchem.ncbi.nlm.nih.gov/, accessed on 12 May 2022). The dataset was cleaned by removing the duplicates. The activity cut-off (IC_50_/EC_50_) value to distinguish between active and inactive compounds was set to 10,000 nM (10 μM). Compounds with activity value ≤ 10 μM were considered as active and were further included into the compound library. 

#### 4.1.1. Molecular Fingerprinting Analysis

A set of active molecules containing isomeric SMILE strings was used as the query for performing 2D molecular similarity/fingerprints (FP) search against PubChem database (https://pubchem.ncbi.nlm.nih.gov/, accessed on 12 May 2022). RDKit (1 March 2022) [30] was used for fingerprint (FP) and molecular similarity analysis. FPs were calculated using molecular similarity assessment based on the Tanimoto coefficient (*T*_c_), which is the ratio of the number of attributes common to both molecules to the total number of features:(1)$$T_{c}\left(a,b\right)=\frac{NC}{N_{a}+N_{b}-N_{c}}$$
where *N* represents the number of attributes in each *a* and *b* molecules, and *c* is the common attribute in *a* and *b*. The range of *T*_c_ varies from 0 to 1, where 0 represents minimal and 1 maximal similarity. 

#### 4.1.2. Compound Filtering

Filtering of the compounds was performed to choose hit-like compounds from the compound list and to remove compounds with undesirable properties. There are numerous computational filters which can be used to identify compounds that may have problems due to assay interference or downstream ADMET properties. The most commonly used of these are physicochemical property calculations based on Lipinski’s descriptors (rule of five); these methods specifically attempt to remove compounds that may lead to low levels of drug absorption and distribution. Molecules which do not possess any of the descriptors were discarded, and molecules that satisfy all the Lipinski rule of five (RO5) criteria were considered for further analysis. Compounds with the same connectivity but different stereochemistry/isotopes were also removed from the dataset. The filtering process also removed salts, invalid molecules, and PAINS (Pan-Assay Interference Compounds) for unspecific and promiscuous compounds because they are frequently identified as hits in a variety of target-based screening [31,32]. Finally, the filtered compounds were noted based on the frequency of their return and subjected to molecular docking calculations. Prior to docking, top hits from virtual screening were converted to isomeric SMILES preserving their stereochemistry. The resulting SMILES strings were employed as input for conformational sampling by ETKDG (Experimental Torsion-angle preference along with basic knowledge-terms and Distance Geometry), a stochastic search method that utilizes distance geometry, together with knowledge derived from experimental crystal structure [33]. Then, energy minimizations employing MMFF94s were performed, with 1000 iterations for each conformation, with maximum number of 10 conformations for each molecule. For each ligand, 10 diverse poses were generated and the best scoring pose was used for molecular docking calculations. All these analyses were performed using Open Source MayaChemTools by utilizing the RDKit library [34].

### 4.2. Protein Preparation

Protein preparation in computational biology is a process in which macromolecular structures are converted into a more suitable form for computational purposes [35]. Prior to docking, refinement of the protein molecule is desirable, hence the protein structure was prepared using the following steps: (i) deletion of heteroatoms, including water molecules, metal ions, and cofactors. (ii) Addition of polar hydrogen bonds followed by removal of atomic clashes. (iii) Gasteiger charges were calculated by means of AutoDockTools (ADT) 1.5.6 tools [36]. Further, Lys, Arg, His, and Cys side chains were protonated, along with deprotonation of Asp and Glu side chains. Missing amino acid residues were added using Šali and Bundell’s Modeller accessed through UCSF Chimera 1.14 [28,37]. For the current study, the structure of LuxP (PDB ID: 1JX6) was obtained from the RCSB PDB, which was determined experimentally and validated through X-ray diffraction method having resolution 1.5 Å and R-value free score 0.239, which is significantly lower than standard value 0.25. 

### 4.3. Molecular Docking and Rescoring Calculation

The top hits compounds from virtual screening were subjected to molecular docking calculations using Smina, a fork of Vina that focuses on improving scoring function and accuracy and is approximately two-fold faster than its predecessor Vina [38]. Here, we used a python script to virtually screen the top hits against the target, LuxP, with the script providing top poses amongst the screened compounds with the lowest binding energy in kcal/mol. 

Binding of ligand/drug molecules to a specific protein site is the key strategy in treatment of many diseases. Attachment of ligand to different proteins may cause side effects and have a higher possibility of toxicity [39]. Binding affinity depends on several features, i.e., hydrogen bond donors and acceptors and hydrophobic or hydrophilic interaction. In this study, we used Smina autobox feature to find the binding site of the desired protein. The autobox_ligand and autobox_add features of Smina use custom scoring function for the prediction of accurate ligand binding sites for protein structure. The predicted binding energy (kcal/mol) indicates how strongly a ligand binds to the protein, which is calculated based on the scoring function used in Vinardo, an extension of Smina scoring function with few additional features that includes a modified term for calculating steric interaction and an estimation of atomic radii [40]. A more negative score indicates stronger binding affinity. It is also possible to apply multiple scoring functions which outperform the classical scoring function such as Vina, Smina, and Vinardo. Therefore, the output from docking calculations were further rescored based on machine-learning scoring function called RF-Score-VS, a novel Random Forest-based scoring function for rescoring outputs from Smina-Vinardo, which has shown great promise by providing much better prediction of measured binding affinity than Smina and Vina [41,42]. Top 5 ranked molecules were then selected for molecular dynamic (MD) simulations.

### 4.4. Molecular Dynamic (MD) Simulations

The AMBER ff14SB force field [43] was used to model the protein and the GAFF force field [44] was used to model the ligands. Such protein–ligand complexes were solvated in a truncated octahedral box of TIP3P water molecules spanning a 12 Å thick buffer, and Na^+^ and Cl^-^ ions were added according to Machado and Pantano [45] to achieve a neutral environment with a salt concentration of 0.15 M. Such structures were then submitted for geometry optimization in the AMBER16 program [46], employing periodic boundary conditions in all directions. For the first 1500 cycles, the complex was restrained (k = 10.0 kcal/ mol Å^2^) and only water molecules were optimized, after which another 2500 cycles of optimization followed where both water molecules and the complex were unrestrained. Optimized systems were gradually heated from 0 to 310 K and equilibrated during 30 ps using NVT conditions, followed by productive and unconstrained MD simulations of 120 ns employing a time step of 2 fs at constant pressure (1 atm) and temperature (310 K), the latter held constant using Langevin thermostat with a collision frequency of 1 ps^−1^. Bonds involving hydrogen atoms were constrained using the SHAKE algorithm [47], while the long-range electrostatic interactions were calculated employing the Particle Mesh Ewald method [48]. The non-bonded interactions were truncated at 11.0 Å. Analysis of the trajectories was performed using the cpptraj module of AmberTools16 [49]. The MD simulation was performed on a single i7-2600 CPU, where wall clock time of about four hours was used per 1 ns simulation.

### 4.5. Binding Free Energy Calculations and Decomposition

The binding free energies, Δ*G*_BIND_, of the simulated complexes were calculated using the MM-GBSA (Molecular Mechanics—Generalized Born Surface Area) protocol [50,51], available as a part of AmberTools16 [46]. Δ*G*_BIND_ is calculated from snapshots of MD trajectory [52] with an estimated standard error of 1–3 kcal/mol [50]. Δ*G*_BIND_ is calculated in the following manner:
Δ*G*_BIND_ = <G_complex_> − <G_protein_> − <G_ligand_>(2)
where the symbol < > represents the average value over 100 snapshots collected from the last 30 ns part of the corresponding MD trajectories (every 150th frame was taken for the calculation). The calculated MM-GBSA binding free energies were decomposed into specific residue contribution on a per-residue basis according to established procedures. This protocol calculates the contributions to Δ*G*_BIND_ arising from each amino acid side chains and identifies the nature of the energy change in terms of interaction and solvation energies [29,53].

## Figures and Tables

**Figure 1 molecules-27-04887-f001:**
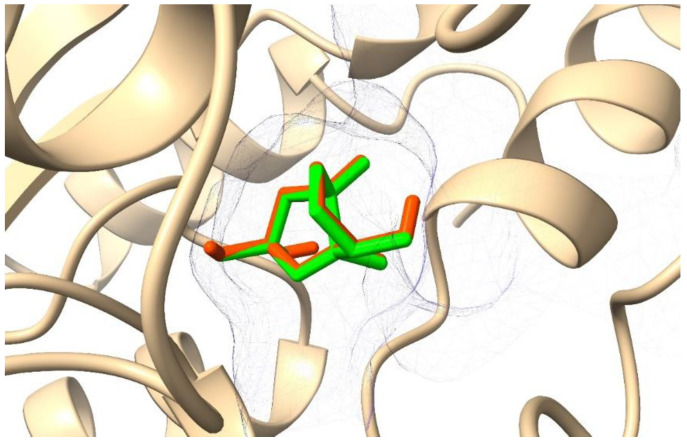
Comparison of the crystallographic (green) and redocked (orange red) AI2 in the binding site of the LuxP protein.

**Figure 2 molecules-27-04887-f002:**
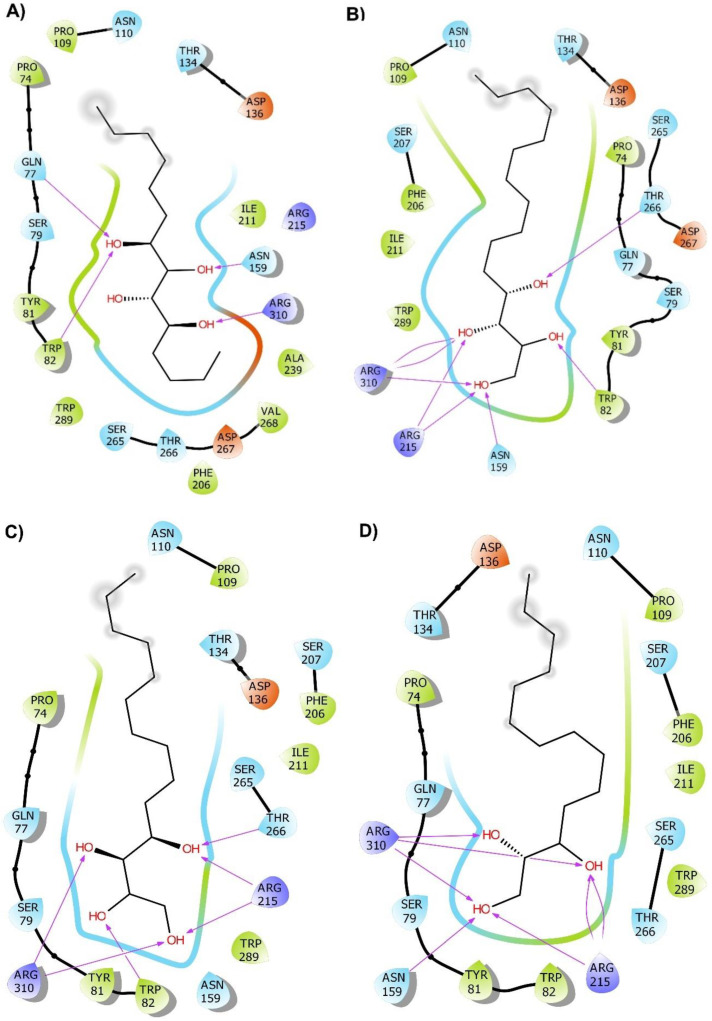
(**A**–**E**) Interaction of the top five potential hits (amino acids are depicted in different colors: green —hydrophobic, blue—polar, orange—negatively charged, purple—positively charged, pink—hydrogen bonds).

**Figure 3 molecules-27-04887-f003:**
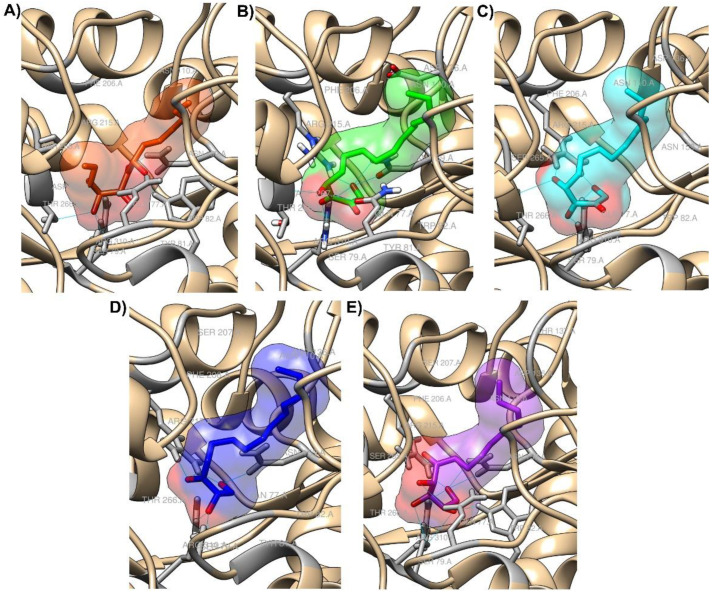
(**A**–**E**) Docked poses of top five potential hits in the active site of LuxP.

**Figure 4 molecules-27-04887-f004:**
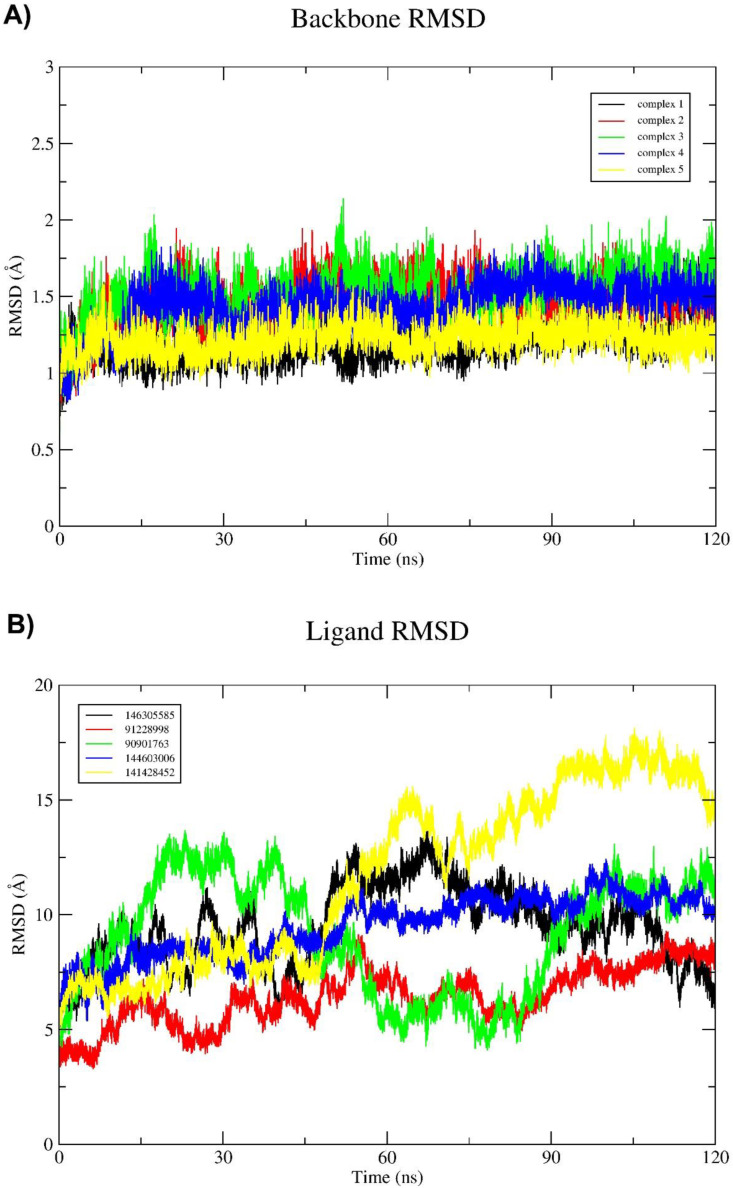
(**A**) Root-mean-square deviation (RMSD) of backbone atoms of all five protein–ligand complexes and (**B**) Ligand RMSD of all five potential inhibitor molecules through time.

**Figure 5 molecules-27-04887-f005:**
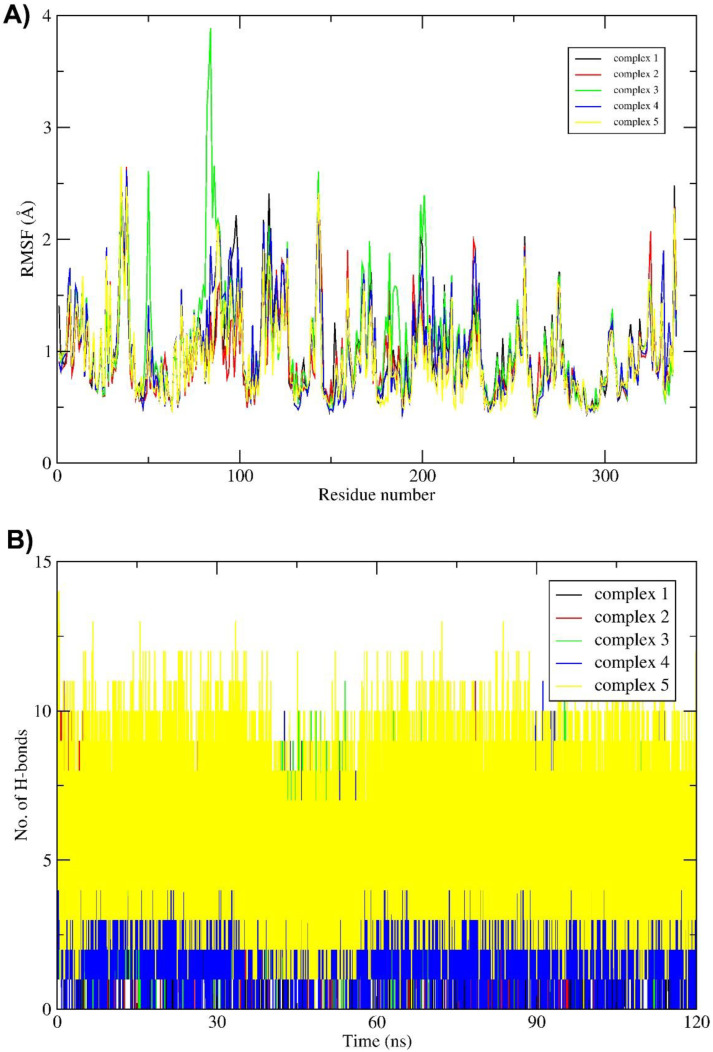
(**A**) Root mean square fluctuation (RMSF) of all non-hydrogen atoms and (**B**) the number of intermolecular hydrogen bonds for the five ligand–receptor complexes.

**Table 1 molecules-27-04887-t001:** Docking energy of the top five hit compounds and their amino acid interactions.

Serial Number	Compound Name (PubChem CID)	Structure of the Compounds	Smina (Binding Energy in kcal/mol)	RF_Score (Binding Affinity in pKd)
1	5,8-dihydroxytetradecane-6,7-dione (146305585)	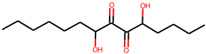	−7.7	6.76
2	1,4-dihydroxypentadecane-2,3-dione (91228998)	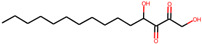	−9.3	7.10
3	1,4-dihydroxytetradecane-2,3-dione(90901763)	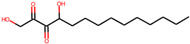	−9.1	6.70
4	1,2-dihydroxytetradecan-3-one(144603006)	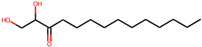	−8.6	6.39
5	(2*R*,3*S*,4*R*)-1,2,3,4-tetrahydroxypentadecan-5-one(141428452)	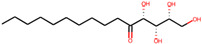	−9.0	6.84

**Table 2 molecules-27-04887-t002:** Average number of intermolecular hydrogen bonds and Δ*G*_BIND_ of the five complexes.

Serial No.	Compound Name (PubChem CID)	H-Bonds		Δ*G*_BIND_ * (kcal/mol)
		Mean	s.d.	
1	5,8-dihydroxytetradecane-6,7-dione (146305585)	2.52	1.28	−38.17
2	1,4-dihydroxypentadecane-2,3-dione (91228998)	3.43	1.32	−44.09
3	1,4-dihydroxytetradecane-2,3-dione (90901763)	3.6	1.36	−40.66
4	1,2-dihydroxytetradecan-3-one (144603006)	2.99	1.51	−41.77
**5**	**(2R,3S,4R)-1,2,3,4-tetrahydroxypentadecan-5-one** **(141428452)**	**5.64**	**1.74**	**−49.43**

* Last 30 ns of the 120 ns simulation using MM-GBSA. Bolded values indicate the compounds that showed the lower Δ*G*_BIND_.

**Table 3 molecules-27-04887-t003:** The top ten contributing amino acid residues of LuxP with top three potential ligand molecules.

Complex 1		Complex 2		Complex 3		Complex 4		Complex 5	
Residue	Δ*G*_BIND_	Residue	∆*G*_BIND_	Residue	∆*G*_BIND_	Residue	∆*G*_BIND_	Residue	∆*G*_BIND_
Val 57	−9.75	Glu 37	−8.74	Glu 37	−6	Asp 36	−8.24	Pro 58	−9.5
Ser 60	−8.5	Ala 54	−6.69	Glu 40	−7.99	Pro 58	−6	Lys 61	−8.49
Leu 96	−8.24	Thr 63	−6.49	Glu 50	−6.25	Ser 60	−5	Gln 64	−5.5
Ile 98	−7.75	Gln 64	−5.99	Val 57	−4.25	Thr 63	−7.74	Leu 96	−7
Thr 199	−7.5	Phe 194	−9.74	Pro 58	−5.49	Leu 96	−4.74	Asn 97	−7.1
His 255	−7.75	Asp 257	−8.5	Leu 59	−6.5	Asn 97	−3.74	Leu 148	−5.25
Asp 257	−9.49	Ala 277	−6.74	Pro 66	−4.49	Ile 98	−4	Pro 256	−8.49
Asp 259	−7.49	Leu 279	−8.5	Ile 98	−9.25	Asn 99	−5.74	MET 311	−7.49
Leu 279	−9.25	Leu 327	−8.99	Asn 99	−6.99	Phe 194	−3.49	Ile 323	−7.49
Leu 355	−9.5	Asp 329	−5.99	Val 223	−5.5	Lys 197	−8.99	GLY 338	−5.75

**Table 4 molecules-27-04887-t004:** Energy contribution to the binding free energy (kcal/mol) for top five potential hits obtained by MM-GBSA approach.

	Complex 1	Complex 2	Complex 3	Complex 4	Complex 5
Δ*E*_vdW_	−42.01	−46.27	−39.85	−44.43	−45.26
Δ*E*_electrostatic_	−27.42	−27.32	−35.39	−19.003	−45.47
Δ*G*_GB_	37.83	36.64	41.02	28.28	48.57
Δ*G*_SA_	−6.56	−7.14	−6.44	−6.62	−7.37

## Data Availability

Available on request.

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
