# Peer review of "In Silico Screening of Quorum Sensing Inhibitor Candidates Obtained by Chemical Similarity Search"

_molecules, 2022, doi:10.3390/molecules27154887_

Round 1
Reviewer 1 Report
Shekarappa et al. “In-silico screening of quorum sensing inhibitor candidates obtained by chemical similarity search” presents a virtual screening of putative inhibitors, where compound filtering, ADMET analysis, docking, and molecular dynamics were performed.
The paper has an organized way of methodology. Validation of docking method by using root mean square deviation values (found 1.176 Å ). The authors used molecular dynamics to validate the docking experiment and study the fluctuation of the inhibitor-target complex. This strategy is quite common as often used to verify the accuracy of the adopted docking methods.
Some points need to address before considering the manuscript in the journal
1. Page 1, line 41, “ It is a well-known fact that molecules that share chemical similarities often share bioactivities as well [19]” is a vague statement. The effectivity of such statement changes with case-by-case study and, to some extent, could be applicable for ligand-based drug design. Even a subtle change in parent scaffold (peptide, carbohydrate, or heterocycle) diminishes the bioactivity (10.1016/j.ejmech.2018.11.030; 10.1016/j.ejmech.2019.04.064). Please elaborate this statement further as this sets the rationality of the manuscript.
2. “We employed the ligand-based drug discovery (LBDD) approach to construct a library of potential AI2 inhibitors, where the molecules were selected based on chemical similarities with known inhibitors [19]”
Authors need to state the chemical similarities more clearly: Wheather they consider 2D descriptors or 3D descriptors?
3. Page 2, line 54, What is a link between the LuxP and AI2? Please provide a brief description to correlate them. If possible, please mention a few sentences in the introduction section of the manuscript.
4. Include a figure showing the 2D structures of compounds in Table 1.
5. Typographical errors Page 2 line 51 “similarities with known inhibitors <SPACE>[19].
6. Subsript of IC50 and EC50 on page 11 line 227.
The manuscript has sufficient elements that makes it fit in the current journal if author agrees to revise the manuscript based on raised points.
Author Response
Comment)
1. Page 1, line 41, “ It is a well-known fact that molecules that share chemical similarities often share bioactivities as well [19]” is a vague statement. The effectivity of such statement changes with case-by-case study and, to some extent, could be applicable for ligand-based drug design. Even a subtle change in parent scaffold (peptide, carbohydrate, or heterocycle) diminishes the bioactivity (10.1016/j.ejmech.2018.11.030; 10.1016/j.ejmech.2019.04.064). Please elaborate this statement further as this sets the rationality of the manuscript.
Reply)
We now revised this part substantially to make the presentation clearer.
Comment) 2. "We employed the ligand-based drug discovery (LBDD) approach to construct a library of potential AI2 inhibitors, where the molecules were selected based on chemical similarities with known inhibitors [19]"
Authors need to state the chemical similarities more clearly: Wheather they consider 2D descriptors or 3D descriptors?
Reply)
We now explicitly stated that they are 2D descriptors.
Comment) 3. Page 2, line 54, What is a link between the LuxP and AI2? Please provide a brief description to correlate them. If possible, please mention a few sentences in the introduction section of the manuscript.
Reply) We now mentioned in the introduction that LuxP is the AI2 receptor of bacteria {\it Vibrio Harvey} which is often used for testing quorum-sensing inhibition activity of AI2.
Comment) 4.Include a figure showing the 2D structures of compounds in Table 1.
Reply) We now included these figures following the request of the reviewer.
Comment) 5. Typographical errors Page 2 line 51 “similarities with known inhibitors [19].
Reply) Unfortunately, we could not find a typo here. We hope any remaining error will be corrected during the typesetting process by the publisher.
Comment) 6. Subscript of IC50 and EC50 on page 11 line 227.
Reply) We now changed them into subscripts.
Comment)
The manuscript has sufficient elements that makes it fit in the current journal if author agrees to revise the manuscript based on raised points.
Reply)
We thank the reviewer for the positive comments. We revised the manuscript according to the suggestion of the reviewer, as detailed above.
Reviewer 2 Report
Dear authors. The paper is interesting and prepared correctly. It refers to the use of the results in a practical way and presents the direction of in vitro research.
I kindly ask you to add information in the introduction as to whether such compounds have already been used in dentistry or whether the research is completely new.
Author Response
Comment)Dear authors. The paper is interesting and prepared correctly. It refers to the use of the results in a practical way and presents the direction of in vitro research.
I kindly ask you to add information in the introduction as to whether such compounds have already been used in dentistry or whether the research is completely new.
Reply)
We thank the reviewer for the positive comments. In fact, D-galactose has been actually used for prevention of dental disease. We now stated this fact in the introduction and added a citation on a relevant patent as a newly added reference [16].
Reviewer 3 Report
The manuscript entitled “In-silico screening of quorum sensing inhibitor candidates obtained by chemical similarity search” is an interesting study done by Shekarappa et al and it is well written. It may be considered for publication in Molecules after the major revision mentioned below.
1. Delete I, we, our throughout the manuscript. For example, we constructed, we used, we searched, we found etc.
2. Line 15 - we assessed whether – This phrase should be changed as it appears to be a hypothesis rather than objective. In the abstract, one line of results (Line 17–18) is insufficient. The significance of the findings should be described in more detail in the abstract. I noticed that the abstract provided more details on the methods than the results. Additionally, make the abstract's conclusion and future directions broader.
3. There are 19 references in the introduction, however 17 of them are outdated and were published more than five years ago. It appears like the introduction section is lacking some crucial details and latest information. More Up-to-date references from the last five years need to be added to this section, which has to be extensively revised.
4. The authors use the data at their fingertips to write the results section well. However, there is room for improvement in the resolution of the figures.
5. The authors should carefully assess how their findings compare to earlier, more recent investigations. I have only noticed one reference in the discussion section which is not appropriate.
6. The details and dates of access to all software platforms must be mentioned in the material and methods section.
7. There must be a part for the conclusion. It is important to emphasise the work's originality. This part should be built on the findings and insights. Finally, this should present an accurate depiction of the study. Future perspectives must be thoroughly covered in the conclusion. The importance of the study must be shown as well.
Author Response
Comment)The manuscript entitled “In-silico screening of quorum sensing inhibitor candidates obtained by chemical similarity search” is an interesting study done by Shekarappa et al and it is well written. It may be considered for publication in Molecules after the major revision mentioned below.
Reply)
We thank the reviewer for the useful comments and the feedbacks.
Comment)
1.Delete I, we, our throughout the manuscript. For example, we constructed, we used, we searched, we found etc.
Reply)
We removed "our", but we could not find "I". Furthermore, we found that it is impossible to completely remove "we". A sentence must have a subject and it is inevitably "we" for most of the sentences. The only solution seems to be to change all the sentences to passive voices, which we think is a bad style. Therefore, we changed only some of the sentences to passive voices, and I hope the reviewer understands.
Comment)
2.Line 15 - we assessed whether – This phrase should be changed as it appears to be a hypothesis rather than objective. In the abstract, one line of results (Line 17–18) is insufficient. The significance of the findings should be described in more detail in the abstract. I noticed that the abstract provided more details on the methods than the results. Additionally, make the abstract's conclusion and future directions broader.
Reply)
We revised the abstract following the suggestion of the reviewer.
Comment)
3.There are 19 references in the introduction, however 17 of them are outdated and were published more than five years ago. It appears like the introduction section is lacking some crucial details and latest information. More Up-to-date references from the last five years need to be added to this section, which has to be extensively revised.
Reply)
We updated the references and extensively revised the introduction as the reviewer requested.
Comment)
4.The authors use the data at their fingertips to write the results section well. However, there is room for improvement in the resolution of the figures.
Reply)
We improved the resolution of Figure 2. The resolutions of the remaining figures seemed reasonable.
Comment)
5.The authors should carefully assess how their findings compare to earlier, more recent investigations. I have only noticed one reference in the discussion section which is not appropriate.
Reply)
Following the suggestion of the reviewer, we now cited recent investigations in the discussion section and made the comparison with the current one. Comment)
6.The details and dates of access to all software platforms must be mentioned in the material and methods section.
Reply)
We already specified the software we used, such as Smina or AMBER16. It is not clear what additional details the reviewer wants us to include. These are softwares installed in our local computers and we accessed them very often, at any time we seemed necessary, and therefore we are not sure it is appropriate to list all the dates we used these softwares, which will be very lengthy and provide no useful information. Instead, we now included the specifications of the hardware and the computational time for MD in the method section.
Comment)
7.There must be a part for the conclusion. It is important to emphasise the work's originality. This part should be built on the findings and insights. Finally, this should present an accurate depiction of the study. Future perspectives must be thoroughly covered in the conclusion. The importance of the study must be shown as well.
Reply)
Upon examining previous publications on Molecules, we found that it is customary to merge conclusion into Discussion section. Therefore, instead of adding Conclusion, we expanded the discussion in order to incorporate the reviewer's suggestions.
Round 2
Reviewer 3 Report
According to the comments, the authors made significant revisions to the manuscript, which might now be accepted for publication in MDPI-Molecules.